# Which Diets Are Effective in Reducing Cardiovascular and Cancer Risk in Women with Obesity? An Integrative Review

**DOI:** 10.3390/nu13103504

**Published:** 2021-10-04

**Authors:** Erika Aparecida Silveira, Priscilla Rayanne E. Silva Noll, Noushin Mohammadifard, Ana Paula Santos Rodrigues, Nizal Sarrafzadegan, Cesar de Oliveira

**Affiliations:** 1Postgraduate Program in Health Sciences, Faculty of Medicine, Federal University of Goiás, Goiânia 74690-900, Brazil; 2Department of Epidemiology & Public Health, Institute of Epidemiology & Health Care, University College London, London WC1E 6BT, UK; c.oliveira@ucl.ac.uk; 3Department of Health and Student Assistance, Instituto Federal Goiano–Campus Ceres, Ceres 76300-0000, Brazil; 4Isfahan Cardiovascular Research Center, Cardiovascular Research Institute, Isfahan University of Medical Sciences, Isfahan, Iran; nmohammadifard@gmail.com; 5Superintendence of Health Surveillance, Department of Goiás State, Goiania 74093-250, Brazil; anapsr@gmail.com; 6School of Population and Public Health, Faculty of Medicine, University of British Columbia, Vancouver, BC V6T 1Z3, Canada

**Keywords:** obesity, cardiovascular and cancer risk, plant-based diet, low-carb diet, intermittent fasting, Mediterranean diet

## Abstract

Women are more affected by obesity than men which increases their risk of cancer and cardiovascular disease (CVD). Therefore, it is important to understand the effectiveness of different types of diet in the context of women’s health. This review aims to summarize the scientific evidence on the effects of different types of diet for women with obesity and their impact on CVD and cancer risk. This review included epidemiological and clinical studies on adult women and different types of diets, such as the Mediterranean (MED) diet, the Traditional Brazilian Diet, the Dietary Approach to Stop Hypertension (DASH), intermittent fasting (IF), calorie (energy) restriction, food re-education, low-carbohydrate diet (LCD) and a very low-carbohydrate diet (VLCD). Our main findings showed that although LCDs, VLCD and IF are difficult to adhere to over an extended period, they can be good options for achieving improvements in body weight and cardiometabolic parameters. MED, DASH and the Traditional Brazilian Diet are based on natural foods and reduced processed foods. These diets have been associated with better women’s health outcomes, including lower risk of CVD and cancer and the prevention and treatment of obesity.

## 1. Introduction

Obesity is a chronic and multifactorial disease which is a risk factor for other conditions such as cardiovascular disease and more than 13 types of cancer. It has also been associated with an increased mortality risk [1]. Obesity and being overweight are more prevalent in women than in men and their occurrence affects two-third of American women. In women, obesity increases the risk of postmenopausal and invasive breast cancers (BC), in addition to the risk of cardiovascular disease [2]. Moreover, in women with a normal body mass index (BMI) but with high body fat, the risk of postmenopausal invasive BC is also elevated [3]. Obesity is also one of the major risk factors for worse COVID-19 outcomes, including a higher risk of mortality [4].

With regard to women’s health, obesity specifically increases infertility rates [5], weight gain during pregnancy with potential postpartum complications for women and children and is associated with miscarriage [6]. Furthermore, obesity in women also impacts negatively on aspects related to the postmenopausal period such as greater weight gain, loss of bone mineral density, reduction of muscle mass, increases in fat-free mass [7,8] and cardiovascular disease [8]. It is worth highlighting that women who underwent bariatric surgery may still be in the overweight and obesity range, and present with micronutrient deficiencies [9,10]. Usually in women, due to their metabolism and body composition, weight loss and control of obesity is more difficult than in men. Obesity treatment is complex and requires a multidisciplinary approach [11]. After a well-elaborated diagnosis, including body composition analyses and a series of biochemical tests, given that individuals with obesity may have anemia or other vitamin and mineral deficiencies, is important to establish the eating habits of these individuals to identify their main eating problems. Therefore, it is essential that a nutritionist makes a diagnosis of their food routine, identifying possible deficits in the intake of some micronutrients, vitamins, and minerals. A multi-disciplinary approach is very important in the treatment of obesity due to its chronic and multifactorial character. However, in such an approach, the role of a nutritionist is pivotal in this type of intervention [11].

Globally, obesity is an important public health problem affecting more women than men, mainly during the climacteric period. Thus, it is important to understand which nutritional and diet therapy treatments are effective in the context of women’s health. Therefore, the aims of this study are: (i) to summarize the scientific evidence on the effects of dietary interventions using different types of diets, on weight loss, reduction of body mass index and abdominal obesity and modification of body composition, (ii) the impact of these diets in the long term on the occurrence of cardiovascular disease and cancer.

## 2. Materials and Methods

This integrative scientific review study includes epidemiological and clinical studies i.e., randomized clinical trials, cohort studies, and systematic reviews. We have evaluated the list of references of systematic reviews to check whether a particular article could be relevant and match our inclusion criteria.

The populations studied were adult women, including pregnant women, or studies that carried out a separate analysis by sex, allowing us to extract the data for women. Studies with adolescents and older adults were not included.

The interventions included were several types of diets, such as: the Mediterranean (MED) diet, the Traditional Brazilian Diet, the Dietary Approach to Stop Hypertension (DASH), intermittent fasting (IF), calorie (energy) restriction, food re-education, low-carb and very low-carb diet also named as very-low-calorie ketogenic diet (VLCKD). In the cohort studies, we have analyzed the impact of food patterns such as the Mediterranean diet during the follow-up period.

There were no language restrictions, and we have considered the last 20 years of publication. We have included the following databases: PubMed, Scopus, Scielo and Web of Science.

As one of the aims was to analyze cardiovascular disease (CVD) and their risk factors, we have used various types of CVDs: heart failure, coronary heart disease, hypertension, cardiomyopathies, cerebrovascular disease, rheumatic heart disease and peripheral vascular disease. With regards to cancer, we used all kinds of malignancies and neoplasms.

BMI is not the most appropriate method to diagnose obesity. However, we have included this measure on the definition of obesity and overweight due to its wide use in epidemiological and clinical studies. We have considered women to be obese when their BMI was >30 kg/m^2^ and overweight when it was >25 kg/m^2^ [12]. We also included abdominal obesity measured by waist circumference (WC), waist-hip ratio (WHR) and body composition with percentage of body fat (%BF). The main outcomes were loss or reduction of weight, BMI, WC or %BF, and we also included cancer and cardiovascular risk factors.

## 3. Impact of Several Types of Diet

### 3.1. Low-Carb and Very Low-Carb Diet

Low-carbohydrate diets emphasize restriction of carbohydrate, which might be replaced with fat and/or protein. However, an agreement on the amount of restriction that determines a low-carbohydrate diet is still a matter of discussion [13,14]. A definition has been proposed based on several publications of experimental studies. Thus, there is a general agreement that “very low-carbohydrate diets” (VLCDs) comprise an intake of less than 20 g of carbohydrates per day. In some studies, the threshold is less than 50 g per day, equivalent to about 10% of total energy intake. For “low-carbohydrate diets” (LCDs), carbohydrate daily intake should reach a maximum of 130 g per day, which is equivalent to less than 26% of total energy [14,15,16,17].

Usually, the LCD interventions in clinical trials are compared to low-fat diets (LFDs) [17,18,19,20,21,22,23,24,25,26,27]. Several meta-analyses have found favorable effect on weight loss for LCDs over LFDs [18,19,20,21,22,23,24], while other meta-analyses found similar results when comparing LCD to LFD or balanced diets [25,26,27]. Additionally, it seems that in the long-term, the effects of LCDs and VLCDs on weight loss may not be superior to more conventional strategies, such as calorie-restricted, low-fat diets and high-carbohydrate low-fat diets [17]. Regarding cardiometabolic risk markers, as observed for weight loss, the results are inconsistent when comparing LCD and VLCD to high-CHO and low-fat diets for high-density lipoprotein (HDL-c), low-density lipoprotein cholesterol (LDL-c), triglyceride (TG), blood pressure, serum glucose and insulin [16,17].

Few studies have been conducted exclusively with women [28,29,30,31] while some have mostly women in their sample i.e., 64% to 89% [32,33,34,35,36,37,38,39]. In addition, there were controversial results for weight loss and obesity outcomes, such as fat mass, waist circumference and visceral fat mass. However, most studies have shown no difference for LCD compared to LFD or commercial diets such as Zone, Learn and Ornish diets [30,31,32,34,35,36,37,38,39]. Out of six studies [31,33,34,35,36,37] with at least one year of follow-up just one observed greater weight and fat mass reduction for LCD compared to LFD (mean difference in change of −3.5 kg [95% CI, −5.6 to −1.4 kg] for weight and −1.5% [CI, −2.6% to −0.4%] for fat mass). There was no significant difference for waist circumference reduction between diets [33] (Table 1).

For cardiometabolic outcomes, LCD and LFD frequently improve blood pressure, lipid, glucose and insulin parameters, but many studies seem to report greater reduction in triglycerides levels and increases in HDL-cholesterol for LCD compared to LFD [31,33,34,35,36,39]. Only one randomized clinical trial investigated a very-low-carbohydrate diet in cancer. A ketogenic diet (<20 g/d of carbohydrate intake) was compared to the American Cancer Society diet (ACS; high-fiber, low-fat) after 12-weeks of follow-up in women with ovarian or endometrial cancer and found greater visceral fat mass and fasting serum insulin reduction for VLCD compared to the ACS diet [29]. The authors also hypothesized that the elevated serum β-hydroxybutyrate could reflect a metabolic environment inhospitable to cancer proliferation [29].

Polycystic ovary syndrome (PCOS) has an important impact on women’s health. The use of LCDs to decrease body weight and facilitate the treatment of infertility in women with obesity with PCOS has been investigated. A systematic review of seven studies found that reduced carbohydrate load can reduce circulating insulin levels, improve hormonal imbalance and resume ovulation to improve pregnancy rates compared to usual diet in overweight and women with obesity [40]. A meta-analysis of eight randomized clinical trials including 327 women with PCOS found that LCDs compared to a control diet significantly improved BMI, lipid levels (total cholesterol and LDL-cholesterol), Homeostatic Model Assessment of Insulin Resistance (HOMA-IR), testosterone (T), sex hormone-binding globulin (SHBG), and follicle-stimulating hormone (FSH), especially in the long-term (> 4 weeks) and when LCD was a low-fat/low CHO diet [41].

Previous meta-analyses of cohort studies have investigated the relationship between carbohydrate intake and all-cause mortality, and also mortality related to other outcomes, such as CVD and cancer, in the general population [42,43,44,45]. Some meta-analyses have found an association of LCD or low-carbohydrate, high-protein diets with increased all-cause mortality [42,43]. A meta-analysis observed that the risk of CVD mortality and incidence were not significantly increased, while another meta-analysis showed that CVD, cerebrovascular and cancer mortality were greater for higher scores of LCD evaluated as quartiles with the highest quartile related to the lowest carbohydrate intake [43]. Two meta-analyses observed that both LCD (HR 1.20; 95% CI 1.09–1.32) and high-carbohydrate diets (HR 1.23; 95% CI 1.11–1.36) were associated with an increased mortality risk [44,45]. In contrast, one of these meta-analyses analysed the source of protein of the diet and it was observed that LCD high in plant-based sources of protein and fat was associated with a lower risk of total (HR 0.89; 95% CI: 0.83–0.94 for highest versus lowest quintile), and CVD mortality (HR 0.82; 95% CI 0.73–0.92 for highest versus lowest quintile) [45]. Thus, the relationship between carbohydrate restriction and all-cause and CVD mortality remains unclear.

In general, the restriction of carbohydrates for women’s health seems to promote similar results in weight loss and body composition compared to LFD. For cardiometabolic outcomes, LCDs and VLCDs may promote similar results as control diets. However, for triglycerides and HDL-cholesterol levels, LCD seem to show better improvements than LFD. Thus, LCDs and VLCD can be good options to achieve improvements in weight and cardiometabolic parameters, but they are difficult adhere to over an extended period. Furthermore, it is reasonable to advise patients regarding the quality of carbohydrate intake, choosing those options associated with reduced cardiometabolic risk, including vegetables, fruits, whole grains, and legumes.

### 3.2. The Mediterranean Diet

Due to synergistic effects between various nutrients, foods, phytonutrients and compounds, focusing on dietary patterns is a better approach to evaluate the relationship between diet and disease [46]. The MED is one of the healthiest dietary pattern in decreasing CVD and cancer risks [47,48] and has been associated with a 12–28% lower risk of CVD, cancer and all-cause mortality [49]. The improvement effect of MED on CVD and cancer may be mediated by reducing oxidative stress, inflammation, obesity indicators, blood pressure, lipid profiles, glucose level and insulin resistance [50].

A recent meta-analysis of 57 RCTs illustrated that MED has reduced the risk of CVD, stroke, angina, pre-diabetes, breast cancer, but not metabolic syndrome (MetS) [51]. A large cross-sectional study conducted with 497,308 European adults (71% women) from ten countries suggested that a higher adherence to MED was associated with a lower WC [52]. In addition, intensive lifestyle modification including MED and exercise in one RCT reduced BMI and altered lipid profiles [53].

Low adherence to MED was more frequent in women with obesity and was associated with an increased asymptomatic atherosclerosis occurrence [54]. Abdominal obesity is a common component of menopausal MetS [55]. The valuable effects of MED in reducing overweight/obesity and abdominal obesity indicators in perimenopausal and postmenopausal women [56,57] was associated with lower estrogen levels [58]. Adiposity, obesity-related breast cancer as well as menopausal status have been associated with the methylation levels of the ZNF577 gene [59,60,61,62]. Furthermore, the MED can moderate the various genes’ methylation like ZNF577 related to noncommunicable diseases such as CVD [63], stroke [64], and cancer [65]. Lorenzo et al. showed that a greater adherence to MED was associated with higher methylation levels of ZNF577 [66]. MED diet also had beneficial changes on weight loss and maintenance, WC, WHR, body fat and some inflammatory markers such as IL-6 and TNF-a after a 4-month period, compared to the United States Department of Agriculture (USDA)’s MyPyramid diet in breastfeeding women [67].

A meta-analysis of six trials reported that compared to low fat diet, MED diet had greater long-term favorable effects on CVD risk factors like BMI, blood pressure, fasting blood glucose, total cholesterol and inflammatory markers such as hs-CRP in individuals with obesity [68]. Another RCT found that the Central European Diet (CED) and MED with calorie restriction (CR) had significantly reduced the effect on body weight, blood pressure and metabolic biomarkers including insulin, HOMA2-IR, total cholesterol, triglyceride, with no difference between diets on postmenopausal women with abdominal obesity and at least one other MetS component [69]. Thus, it was concluded that CR, irrespective of their macronutrient compositions, could improve obesity and other CVD risk factors [69]. Moreover, a recent meta-analysis of thirty RCTs with moderate quality evidence in primary prevention of CVD risk factors and low-quality evidence of little or no effect in secondary prevention indicated a significant blood pressure reduction in MED compared to no diet as well as reductions in triglycerides and LDL-cholesterol levels compared to another diet in primary prevention. However, there were no changes in secondary prevention. Thus, it was concluded that there was uncertainty related to the effect of MED on preventing CVD [70] (Table 2).

The favorable effects of MED are likely due to synergistic interactions among diverse elements of this diet rather than specific food groups [71]. Some potential mechanisms which explain the cardio-protective role and obesity prevention of MED are its beneficial effects on insulin resistance, endothelium-dependent vasoreactivity, oxidation and inflammation biomarkers [72,73].

Current evidence has shown a protective effect of MED in the risk of cardiovascular disease and cancer in women with obesity in different life cycles. This could be attributed to the various genes’ methylation related to NCDs such as CVD [63] and cancer [65]. The MED also had beneficial changes in weight loss and maintenance, reduction of body fat and inflammatory factors [67]. However, there is some controversy about the effect of MED on the secondary prevention of CVD [70].

### 3.3. The Traditional Brazilian Diet

A diet pattern named the Traditional Brazilian Diet [74,75,76,77] was tested in a randomized clinical trial as a treatment in individuals with class II/III obesity (BMI = 35 kg/m^2^), in which more than 85% of the sample where women. We have decided to include this diet in this review due to its features, such as being a kind of plant-based and reduced ultra-processed foods. It is a healthy diet pattern which can be easily incorporated in eating habits due to its common food components such as rice, beans, fruits, and vegetables largely consumed in many cultures. This diet pattern does not include food such as nuts, olive oil, seafood and wine, which can be difficult to find in most countries or are expensive for people who live in low- or middle-income countries. A comparative analysis of the MED and Traditional Brazilian diets can be found in a previous publication [77].

To better characterize the Traditional Brazilian Diet, consider a dinner plate and divide its half into three parts that will be one part rice, one part beans and one part lean red or white meat. The other half of the plate will be filled with boiled or raw vegetables in the form of salad, grilled or baked culinary preparations [74,75,76,77].

Cardiovascular risk is a public health issue worldwide, which increases the risk of mortality mainly in postmenopausal women. Therefore, it is important to reduce this risk factor. The Traditional Brazilian Diet intervention was effective in decreasing some cardiometabolic risk parameters in individuals with severe obesity, mainly LDL-cholesterol, HbA1c, triglycerides and triglycerides/HDL ratio [77]. This diet pattern had not been analyzed in terms of its impact on cancer. Overall, women have a higher prevalence of anxiety and depression [74]. In the abovementioned RCT, in which 85% of the sample were women, the Traditional Brazilian Diet showed a significant reduction of 46% of anxiety symptoms, 50% of depression and 67% of both anxiety and depression [74]. After a 12-week follow-up, those participants with severe obesity had a mean weight reduction of −2.83 ± 5.79 kg [74]. Even a modest weight loss can lead to health benefits. This diet pattern has been shown to be more effective in reducing other risk factors that affect women with severe obesity than weight loss itself. The Traditional Brazilian Diet can potentially be a good option to treat women with obesity when the objectives are to reduce cardiometabolic risk, depression, and anxiety symptoms.

### 3.4. DASH

The DASH is a dietary pattern originally developed to treat hypertension without medication by the United States National Institutes of Health (NIH). It is characterized for eating a high number of vegetables which will result in high levels of potassium, magnesium, and calcium, and limits the consumption of macro- and micronutrients that have been pointed out as a risk factor for hypertension, as total and saturated fat, cholesterol, and sodium. The DASH diet has mainly fruit and vegetables, low-fat dairy foods, whole grains, nuts, and legumes and low consume of red and processed meats, sweets, and sugar-sweetened beverages [78,79].

There is relevant evidence of DASH on prevention and treatment of hypertension and cardiovascular risk, such as reducing body weight, LDL-C and insulin. DASH is recommended by the international diabetes and cardiovascular clinical association guidelines [80,81,82,83].

Some studies that have evaluated DASH diet in overweight and individuals with obesity do not present data stratified by sex [84,85,86]. In a meta-analysis of 13 randomized controlled clinical trials, which included 2292 overweight and adults with obesity [87], only three studies presented women’s data separately (*n* = 213) [88,89,90]. The main findings showed a significant association between DASH diet and BMI reduction, compared to the control diet [87]. Some studies have been conducted only with women [91,92,93,94], as shown in Table 3. Four randomized clinical trials have assessed the consumption of calorie-restricted DASH diet compared to a calorie-restricted control diet in overweight and women with obesity with polycystic ovary syndrome for 8–12 weeks [91,92,93,94]. The calorie-restricted DASH diet resulted in greater body weight, BMI [91,92,93,94], fat mass, WC [91] and hip [94] circumferences reduction. These clinical trials also evaluated cardiometabolic outcomes and found a greater reduction in insulin levels, HOMA-IR score, triglycerides, VLDL-cholesterol, total antioxidant capacity, total glutathione, and nitric oxide in overweight and women with obesity with polycystic ovary syndrome after a calorie-restricted DASH diet [91,92,93,94] (Table 3).

A cohort study with 1760 pregnant women found no association between adherence to the DASH diet during early pregnancy, compared to a control diet, and pregnancy outcomes or complications. However, adherence to the DASH diet was unexpectedly associated with greater gestational weight gain in women with obesity before pregnancy [95]. The Women’s Health Initiative, a cohort study with 93,122 postmenopausal women, found no association between a higher DASH diet score and cardiovascular mortality. They found an association between higher DASH diet quintiles and lower BMI and lower waist-to-hip ratio [96].

The DASH diet seems to be effective to reduce and control cardiovascular diseases, as well as reducing weight in women with obesity [87,91,92,93,94,96], except for pregnant women with obesity [95]. The DASH diet may be relevant to treat obesity, mainly when the focus is to reduce cardiometabolic risk; however, the impact of this eating pattern on cancer has not been analysed.

### 3.5. Intermittent Fasting

The Intermittent fasting (IF) diet has become increasingly popular for weight reduction in the past two decades. It generally involves a calorie restriction of 75–90% with 1–3 days/week of fasting [97]. The IF includes alternate day fasting (ADF) (25% of energy needs on fasting days; 125% of energy needs on alternating “feast days”) and periodic fasting (PF). There have been some controversies in the effect of IF and CR diets in human studies [98,99,100,101,102,103,104,105,106,107].

A RCT indicated similar effects of ADF and CR diets on reducing CVD risk factors, weight loss and weight maintenance after one year [98]. Another RCT showed circulating leptin reduction and increased free fat mass (FFM) to total mass ratio without affecting the visceral adipose tissue to subcutaneous adipose tissue (VAT: SAT) ratio and other adipokines during a 24-week intervention. However, HOMA-IR had a greater reduction in ADF compared to the CR group [99]. Contrary to the assumption of easier compliance in ADF, this study showed less sustainability of ADF due to dissatisfaction of subjects with long-term ADF compared with CR [98]. A RCT in overweight and women with obesity also revealed similar improvements in body composition through these two interventions [100].

Time restricted eating (TRE) has been shown to result in good weight management and cardiometabolic beneficial effects including reduction in body weight, VAT, total fat mass, fasting blood glucose (FBG), impaired glucose tolerance, insulin resistance, dyslipidaemia, hypertension, appetite and inflammatory markers [101,102,103]. Especially, increasing the time of fasting from 12 to 14 h per day could produce more improvements in weight loss and FBG [102].

An RCT among overweight and with obesity East Asians in Hawaii showed a significant reduction in obesity indicators, including body weight, BMI, WC, VAT, SAT, body fat percent and total fat mass. However, there was no reduction of the VAT: SAT after IF: MED (2 consecutive days with 70% energy restriction: 5 days euenergetic MED) compared to euenergetic DASH diet after 12 weeks. There was also a decrease in the total lean body mass and muscle [104]. A recent review by Dong et al. showed that both IF and CR diets could reduce CVD risk factors including hypertension, insulin resistance and dyslipidaemia. In addition, IF was linked with CVD events in cardiac patients and weight reduction in individuals with obesity. The potential mechanisms for CVD prevention of IF consist of improving oxidative stress, promoting ketogenesis and a close relationship with the circadian rhythm hypothesis. Due to the time restricted nature of fasting, IF has better adherence and hence increased chance of more weight reduction in individuals with obesity than CR diet [105]. A meta-analysis of seven RCTs among 269 subjects demonstrated that ADF for at least one month could reduce body weight, BMI, fat mass, lean mass, blood pressure, and improve cardiometabolic risk factors including total cholesterol, LDL-cholesterol and triglycerides levels compared with the control group in normal weight and participants with obesity. For the first time, this meta-analysis illustrated that ADF could have greater beneficial effects than CR diet in normal and overweight individuals. ADF plus physical activity produced superior cardiometabolic improvements and weight related indicators such as the least decrease in lean mass compared with ADF alone [106]. Some studies on women showed the improvement effects of IF on obesity and CVD risk factors. A 24-week RCT with women with obesity showed body weight, LDL-cholesterol and triglycerides reductions of 7%, 10% and 17%, respectively [107].

A combination of IF with CR diet (IFCR) showed a stronger effect in reducing weight and CVD risk factors compared to each intervention alone [108]. In addition, a RCT in women with obesity showed that IFCR in its liquid (IFCR-L) had a stronger effect in reducing body weight, BMI, fat mass VAT, glucose, insulin, heart rate, total cholesterol, triglyceride and LDL-cholesterol as well as LDL-cholesterol particle size, but no changes on fat-free mass, SAT, blood pressure and CRP compared to with normal food (IFCR-F) in weight loss period. The greater weight loss and hence other better cardioprotective effect of the IFCR-L intervention is likely to be attributed to its better dietary adherence [108]. 

Different types of IF diet can have a reduction effect on obesity and body composition. However, it can be complicated to sustain the use of IF for a prolonged period. In addition, IF may reduce various CVD risk factors in cardiac patients. We summarized the main evidence for IF in studies conducted in women in Table 4.

### 3.6. General Healthy Diet and/or Food Re-Education

Healthy eating is defined as a diet capable of promoting health and preventing diseases, reducing the risk of being overweight/with obesity and to develop CVD and cancer [109]. General public health recommendations on healthy eating to prevent NCDs include frequent consumption of fruits, vegetables and legumes, oilseeds and whole grains and limited intake of saturated fat, trans fat, sugar and salt [109,110,111,112]. As a general recommendation for the entire population, the combination of foods and meals should also consider the traditional/cultural dietary patterns of each population and sustainable food systems. In other words, involving a diversified diet, considering cultural traditions, geographical and environmental aspects [112]. A systematic review showed an association between diet quality indices and lower percentage of body fat, lower BMI and abdominal obesity, and lower weight gain in adults of both sexes [113].

Regarding meals, a 12-week randomized clinical trial with 93 overweight and women with obesity with metabolic syndrome compared the weight loss in two isocaloric diets (1400 kcal): one diet with high caloric intake during breakfast (700 kcal) (BF) and the other diet with high caloric intake at dinner (700 kcal) (D). The BF group showed greater weight loss, waist circumference, serum ghrelin and lipids, and insulin resistance indices reduction than the D group [114]. Two other randomized controlled trials evaluated the association between snack and weight loss in women and their results indicated that a reduced-calorie diet containing snacks may contribute to weight loss, depending on whether snacks consist of healthy foods, such as fruits, vegetables, and dark chocolate or reflect unhealthy eating habits and may in fact contribute to weight gain [115,116] (Table 5).

Most studies have assessed women with obesity during their reproductive period, including pregnancy. During this period, a healthy and balanced diet, associated with nutritional education provided by a nutritionist/dietitian, has been shown to be crucial to prevent excessive weight gain and postpartum weight retention in women [117,118,119,120,121,122]. It also is a protective factor for the occurrence of gestational diabetes and hypertension, and pre-eclampsia [119,123]. 

A randomized clinical trial [121] and a systematic review [119] demonstrated that sugary food consumption was a risk factor for greater gestational weight gain in a cohort study of 46,262 pregnant women [122]. Women eating healthier diets, assessed by the Healthy Eating Index, have a lower risk of cancer mortality, according to a meta-analysis of cohort studies that evaluated 638,770 adult women [124].

We have used the NOVA classification to define healthy eating. That is, the major consumption of fresh and minimally processed foods, with the contribution of culinary ingredients and processed foods, characterizing culinary preparations [125,126,127,128,129]. Regarding NOVA, we did not find randomized controlled trials evaluating the impact of food consumption on women with obesity. A recent meta-analysis presented data from two studies showing a significant association between the consumption of ultra-processed foods and greater gestational weight gain in pregnant women [130].

Public health policies and programs to support population to promote a healthy food environment are important instruments for the prevention of obesity and other NCDs and require the involvement of government, the public and private sectors [131]. Food guides with graphic representations of the diet [132] and healthy eating recommendations are part of these initiatives and are relevant guidelines for the general population to adopt new healthy eating habits [132,133,134,135,136,137,138,139,140,141]. Another example is the program 5-a-day, which is a campaign to help people ensure that they eat five portions of fruit and vegetables a day [142,143,144,145]. In view of the prominence of this theme for health promotion and disease prevention, as well as the lack of specific recommendations targeted at women, it is relevant to have programs that focus their approach on specific recommendations for them.

## 4. Conclusions

The Med [146], DASH [80,87,147] and the Traditional Brazilian Diet [74,75,76,77,148] have in common the feature of being varieties of “plant-based diet” with the incorporation of natural foods and a reduction of ultra-processed foods. These diets have shown good results for women’s health through the prevention and treatment of obesity in their different life cycles [149], and there is also evidence of a reduction in the risk of cardiovascular disease and cancer in individuals with obesity. In addition, these diets promote a reduction in low-grade inflammation that affects individuals with obesity. Therefore, it is worth following the campaigns of “unpack less and peel more” as ways to reduce the consumption of industrialized products, especially those with high concentrations of sugar, sodium and sugary drinks, and to increase the consumption of natural foods such as fruits, vegetables, and whole grains.

LCDs, VLCD diets [31,33,34,35,36,37] and IF [107,108] seem to promote interesting results with regards to weight control and reduced CVD and cancer risk in women with obesity. However, such diets may be difficult to adhere to over an extended period. From a dietary point of view, we must consider that a healthy diet should be learned and incorporated into the daily routine and not only for some periods with a focus only on weight loss. This type of nutritional treatment, which relies mostly on the adoption of a healthier dietary pattern and food education, is the best approach to prevent and treat overweight and obesity in women and to reduce CVD and cancer risk.

## Figures and Tables

**Table 1 nutrients-13-03504-t001:** Summary of studies with low-carbohydrate diets for weight loss, cardiometabolic and cancer outcomes.

First Author, Year, Country	Design, Follow-Up and Population	Low-Carbohydrate Diet	Comparators/Control	Obesity Outcomes	Cardiometabolic and Cancer Outcomes
Bhrem et al., 2003 USA	RCT 6 months follow-up 53 women ≥ 18 years BMI 30–35 kg/m^2^	Ad libitum very low-carbohydrate diet (LCD) of maximum intake of 20 g/d of CHO for 2 weeks, followed by an increase to 40–60 g/d *n* = 22	Calorie-restricted, moderately low-fat diet (LFD) (55% carbohydrate, 15% protein, and 30% fat). Calorie prescription based on the Harris–Benedict equation *n* = 20	↓ weight for LCD group (8.5 ± 1.0 kg) vs. LFD group (3.9 ± 1.0 kg) (*p* < 0.001) ↓ fat mass and lean mass for LCD group vs. LFD group	↔ blood pressure, ↓ total cholesterol, LDL-c, TG, glucose, and insulin and ↓ HDL-c (no significant difference between groups)
Foster et al., 2003 USA	RCT 12 months follow-up 63 adults, 685 women 44.0 ± 9.4 years BMI: 33.9 ± 3.8 kg/m^2^	LCD: CHO intake <20 g/d for the first 2 weeks, with a gradual increase until stable and desired weight was achieved. Instructed to follow the Atkins diet *n* = 30	LFD: 60% of total energy as CHO, 20% as fat and 10% as protein. Energy intake limited to 5021–6276 kJ (1200−1500 kcal/d) for women and 6276–7531 kJ (1500−1800 kcal/d) for men *n* = 33	↓ weight: greater for LCD at 3 months (mean [−6.8 ± 5.0 vs. −2.7 ± 3.7% of body weight; *p* = 0.001) and 6 months (−7.0 ± 6.5 vs. −3.2 ± 5.6% of body weight, *p* = 0.02), but with no difference at 12 months (−4.4 ± 6.7 vs. −2.5 ± 6.3% of body weight, *p* = 0.26).	↔ systolic blood pressure ↓ diastolic blood pressure for both diets, no difference between them ↔ area under the glucose curve ↓ area under the insulin curve for both diets, no difference between them ↔ LDL-c ↓ TG and ↑ HDL-c for LCD vs. LFD throughout the study
Yancy et al., 2004 USA	RCT 6 months follow-up 120 overweight, hyperlipidemic volunteers from the community 18–65 years, 77% women BMI: 30–60 kg/m^2^	LCD: CHO intake limited to <20 g/d. Increase of 5 g/week until body weight was maintained	LFD: <30% of total energy as fat, <10% SFA and <300 mg cholesterol daily	↓ weight greater for LCD vs. LFD (mean change, −12.9% vs. −6.7%; *p* < 0.001) ↓ fat mass (−9.4 kg for LCD, −4.8 kg for LFD, no difference between groups) ↓ fat-free mass (−3.3 kg for LCD, −2.4 kg for LFD, no difference between groups)	↓ TG greater for LCD vs. LFD (−0.84 mmol/L vs. −0.31 mmol/L [−74.2 mg/dL vs. −27.9 mg/dL]; *p* < 0.004) ↑ HDL-c greater for LCD vs. LFD (0.14 mmol/L vs. −0.04 mmol/L [5.5 mg/dL vs. −1.6 mg/dL]; *p* < 0.001) ↔ LDL-c
Gardner et al., 2007 USA	RCT 12 months follow-up 153 overweight/ with obesity nondiabetic, premenopausal women 25–50 years BMI: 27–40 kg/m^2^	Atkins diet: CHO < 20 g/d or less in the induction phase (2−3 months), and ≤50 g/d or less for the subsequent ongoing weight loss phase *n* = 77	Zone diet: 40%–30%–30% distribution of CHO, protein, and fat. *n* = 79 Learn diet: 55% to 60% energy from carbohydrate and less than 10%energy from saturated fat, caloric restriction *n* = 79 Ornish diet: <10% of total energy from fat *n* = 76	↓ weight: −4.7 kg (95%CI, −6.3 to −3.1 kg) for Atkins, −1.6 kg (95% CI, −2.8 to −0.4 kg) for Zone, −2.2 kg (95% CI, −3.6 to −0.8 kg) for LEARN, and −2.6 kg (95% CI, −3.8 to −1.3 kg) for Ornish and was significantly different for Atkins vs. Zone	↓ HDL-c for Atkins vs. Ornish ↓ TG for Atkins vs. Zone ↓ systolic blood pressure for Atkins vs. the other diets ↓ diastolic blood pressure for Atkins vs. Ornish ↔ fasting insulin or fasting glucose
Morgan et al., 2008 UK	RCT 6 months follow-up overweight and with obesity men and women 18–65 years, 70% women BMI: 27–40 kg/m^2^	LCD prescribed as Atkins diet after Dr Atkins’ New Diet Revolution *n* = 57	LFD: Rosemary Conely ‘Eat yourself slim’ Diet and fitness plan-an energy-controlled and low-fat healthy eating diet and group exercise class *n* = 58 Weight Watchers Pure Points programme: an energy-controlled low-fat healthy eating diet *n* = 58 Slim-fast diet: a low-fat meal replacement approach (up to two meal replacements) *n*= 59 Control group: subjects were asked to maintain their current diet and exercise pattern *n* = 61	↓ weight for all dieting groups (5–9 kg at 6 months) but no significant difference between diets	↓ LDL-c for the Weight Watchers and Rosemary Conley diets (both −12.2%, *p* < 0.01) ↑ LDL particle size for the Atkins and Weight Watchers diets ↓ TG for the Atkins and Weight Watchers diets (–38.2% and –22.6%, *p* < 0.01) ↓ fasting insulin for all diets with no difference between them ↔ fasting glucose
Brinkworth et al., 2009 Australia	RCT 12 months follow-up 69 adults with abdominal obesity and at least one additional metabolic syndrome risk factor 18–65 years 64% women	LCD: 4% of total energy as CHO, 35% as protein, 61% fat (20% SFA). Restriction of CHO to <20 g/d the first 2 months and then <40 g/d for the remainder of the intervention period *n* = 33	LFD: 30% as fat (8% or 10 g/d as SFA), 46% as CHO and 24% as protein *n* = 36	↓ weight and body fat in both groups (no difference between groups)	↓ blood pressure, fasting glucose, insulin, insulin resistance, and C-reactive protein in both groups (no difference between groups) ↓ TG, ↑ HDL and ↑ LDL for VLCD vs. LFD
Foster et al., 2010 USA	RCT 24 months follow-up 307 adults, 68% women 18–65 years BMI: 30–40 kg/m^2^	LCD: <20 g CHO for the first 3 months, thereafter, a gradual increase in CHO intake (5 g/d per week). Participants followed guidelines as described in Dr. Atkins’ New Diet Revolution *n* = 153	LFD: 55% of energy from CHO, 30% from fat and 1% from protein. Energy intake was limited to 5021–6276 kJ (1200−1500 kcal/d) for women and 6276–7531 kJ (1500−1800 kcal/d) for men *n* = 154	↓ weight: approximately −11 kg (11%) at 1 year and 7 kg (7%) at 2 years, with no differences in weight, body composition, or bone mineral density between the groups at any time point	↓ systolic blood pressure, TG, LDL, VLDL, but with no difference between groups at 2 years ↑ HDL-c at all time points, approximating a 23% increase at 2 years for LCD ↓ diastolic blood pressure at 3 months, 6 months and 2 years for LCD
Lim et al., 2010 Australia	RCT 3 months of intervention 12 months of follow-up 20–65 years, 80% women BMI: 28–40 kg/m^2^	LCD: 4% of energy as CHO, 35% as protein and 60% fat (20% SFA) *n* = 27	LFD: 70% of energy as CHO, 20% protein and 10% fat (3% SFA) *n* = 28 High unsaturated fat diet (HUF): 20% energy as protein, 30% fat, 6% saturated fat, 8% polyunsaturated fat, 50% carbohydrate *n* = 27 No intervention *n* = 22	↓ weight: −3.0 ± 0.2 kg for LCD, −2.0 ± 0.1 kg for LFD, −3.7 ± 0.01 kg for HUF and 0.8 ± 0.5 kg for controls (significant difference for all diets vs. control)	↓ systolic and diastolic blood pressure for LCD, LFD and HUF vs. control At 3 months: ↑ HDL, ↓ TG, ↑ homocysteine for LCD compared to the other diets, but with no difference after the 12 months of follow-up ↔ fasting insulin or fasting glucose
Bazzano et al., 2014 USA	RCT 12 months follow-up 148 men and women (89%) without clinical cardiovascular disease and diabetes	LCD: CHO intake <40 g/d. Ad libitum diet with no set energy goal *n* = 73	LFD: <30% of total fat intake as fat, and <7% as SFA. 55% of total energy intake as CHO. No energy restriction *n* = 75	↓ weight for LCD vs. LFD (mean difference in change, −3.5 kg [95% CI, −5.6 to −1.4 kg]; *p* < 0.001) ↓ fat mass for LCD vs. LFD (mean difference in change, −1.5% [CI, −2.6% to −0.4%]; *p* = 0.011) ↓ WC for both groups, with no difference between them	↓ total to HDL-c ratio for LCD vs. LFD (mean difference in change, −0.44 [CI, −0.71 to −0.16]; *p* = 0.002) ↓ TG for LCD vs. LFD (mean difference in change, −0.16 mmol/L [−14.1 mg/dL] [CI, −0.31 to −0.01 mmol/L {−27.4 to −0.8 mg/dL}]; *p* = 0.038) ↑ HDL-c for LCD vs. LFD (mean difference in change, 0.18 mmol/L [7.0 mg/dL] [CI, 0.08 to 0.28 mmol/L {3.0 to 11.0 mg/dL}]; *p* < 0.001) ↔ systolic and diastolic blood pressures ↔ plasma glucose ↓ CRP for LCD vs. LFD (mean difference in change −15.2 nmol/L [CI, −27.6 to −1.9 nmol/L]) ↓ serum levels of insulin and creatinine for both groups, with no difference between them
Cohen et al., 2018 USA	RCT 12 weeks follow-up 45 women with ovarian or endometrial cancer ≥19 years BMI: ≥18.5 kg/m^2^	KD (70:25:5 energy from fat, protein, and CHO) *n* = 25	American Cancer Society diet (ACS; high-fiber, low-fat) *n* = 20	↓ android fat mass in the intervention group (KD: −0.7 vs. ACS: −0.45 kg, *p* < 0.05) ↓ visceral fat in the intervention group (KD: –21.2% vs. ACS: –4.6%, *p* < 0.05).	↓ insulin (KD: 6.7 vs. ACS: 11.2 μU/mL, *p* < 0.01) ↓ C-peptide (KD: 2.0 vs. ACS: 3.0 ng/mL, *p* < 0.01) ↑ β-hydroxybutyrate (KD: 0.91 vs. ACS: 0.25= mmol/L, *p* < 0.001) ↔ Fasting glucose, IGF-1, IGFBP-1
Barry et al., 2021 USA	Non-randomized clinical trial 15 weeks follow-up 50 adults 82% women 45–75 years BMI: ≥ 25 kg/m^2^	Low-Carbohydrate High-Fat Diet (LCHF): 5% CHO, 30% proteins, 65% fats *n* = 32	LFD: 63% CHO, 13–23% proteins, 10–25% fats, 1220–1660 kcal *n* = 18	↓ total body weight for both groups (LCHF: −6.1 ± 5.2 kg, LF: −3.1 ± 4.5 kg) ↓ fat mass for both groups (LCHF: −5.4 ± 3.6 kg, LF: −3.0 ± 3.2 kg) ↔ lean mass changes for both groups ↓ visceral fat reduction for LCHF vs. LF (15.6 ± 22.2% vs. 8.3 ± 8.1%, *p* < 0.01) ↓ trunk, android and gynoid lean mass for LCHF vs. LFD Subgroup analysis of insulin-resistant participants: ↓ android and visceral fat for LCHF vs. LFD in insulin-resistant participants	↓ HOMA-IR for both groups with no difference between groups
Hwang et al., 2021 USA	RCT 6 weeks follow-up 21 healthy women with obesity 33 ± 2 years BMI: 33.0 ± 0.6 kg/m^2^	LCD without caloric restriction: 10% CHO, 60–62% fat, 28–30% protein *n* = 9	LCD with caloric restriction (LCD-CR) of 500 calories/day *n* = 12	↓ body weight, BMI and % body fat in both interventions, no difference between groups	↔ flow mediated dilation, nitro-glycerine mediated dilation, serum nitrate/nitrite levels LCD: ↑ flow induced dilation (FID) by 11% vs. baseline, and endothelial nitric oxide synthase inhibitor (L-NAME) decreased overall FID at week 6 by 20% LCD-CR: ↔ FID, L-NAME decreased overall FID by 19% ↓ diastolic blood pressure and TG in both interventions, no difference between groups

CHO: carbohydrate, CRP: C-reactive protein HDL-c: high density lipoprotein cholesterol, KD: ketogenic diet, LCD: low-carbohydrate diet, LDL-c: low-density lipoprotein cholesterol LFD: low-fat diet, RCT: randomized clinical trial, SFA: saturated fatty acids, TG: triglycerides, USA: United States of America, UK: United Kingdom, WC: waist circumference; ↔ no significant change, ↑ significant increase, ↓ significant reduction.

**Table 2 nutrients-13-03504-t002:** Summary of studies with the Mediterranean diet (MED) for weight loss, obesity and cardiometabolic and cancer outcomes.

First Author, Year, Country	Design and Population	Intervention	Comparator/ Control Group	Obesity Outcomes	Cardiometabolic and Cancer Outcomes
Andreoli et al., 2008 Italy	Clinical trial (before-after design) 4 months follow-up N = 60 women BMI: 25.0–47.8 kg/m^2^	MHMD and exercise program	Before intervention	↓ Significant weight reduction (Before vs. after: Mean 80.4 ± 15.8 vs. 75.2 ± 14.7 kg) ↓ Significant BMI reduction (Before vs. after: Mean 30.7 ± 6.0 vs. 28.7 ± 5.6 kg/m^2^) ↓ Significant FM reduction (Before vs. after: Mean 29.5 ± 10.3 vs. 26.2 ± 10.0 kg)	↓ Significant TC reduction (Before vs. after: Mean 207.4 ± 3.7 vs. 195.6 ± 14.9 mg/dL) ↓ Significant LDL-C reduction (Before vs. after: Mean 111.2 ± 23.5 vs. 106.2 ± 20.2 mg/dL) ↑ Significant HDL-C increase (Before vs. after: Mean 55.2 ± 4.2 vs. 56.1 ± 9.6 mg/dL) ↓ Significant TG reduction (Before vs. after: Mean 180.7 ± 10.6 vs. 175.4 ± 9.7 mg/dL) ↓ FBG (Before vs. after: Mean 92.5 ± 14.5 vs. 89.4 ± 11.4 mg/dL, no difference) ↓ SBP (Before vs. after: Mean 136.9 ± 13.1 vs. 135.4 ± 10.4 mmHg, no difference) ↓ DBP (Before vs. after: Mean 74.4 ± 7.1 vs. 82.9 ± 5.8 mmHg, no difference)
Buscemi et al., 2009 Italy	RCT 2 months follow-up N = 20 overweight-with obesity women BMI: 27–39 kg/m^2^ otherwise healthy, non-smoking and non-pregnant	Hypocaloric MED (M)	Very-low-carbohydrate hypocaloric (A)	↓ weight was significantly higher in A than M group (A: 8.8 ± 0.9 vs. M: 5.9 ± 0.8) ↓ WC was significant in both groups but no difference between two groups No significant reduction in body fat and WHR in both groups	↓ Significant SBP reduction only was higher in A group than M group No significant reduction in DBP, HDL-C, TG, uric acid, FBG and adiponectin in both groups ↓ TC, LDL-C, INS and HOMA-I were significantly higher in A than M group No significant reduction in tumour necrosis factor-a in both groups
Romaguera et al., 2009 10 European countries **	Cross-sectional 497,308 men and women (70.7%) aged 25–70 years	Adherence to mMDS	-	BMI and WC changes:	Change in BMI per one unit mMDS: β (95% CI): −0.01 (−0.04, 0.02) Change in WC per one unit mMDS: β (95% CI): −0.12 (−0.17,−0.08)
Nordmann et al., 2011	Meta-analysis 6–24 months follow-up 6 RCT N = 2650 individuals, 50% women BMI = 29–35 kg/m^2^	MED diet	Low fat diet	↓ Weight was greater in MED vs. Low fat diet: Mean difference (95%CI): −2.24 (−3.85, −0.63) kg ↓ BMI was greater in MED vs. Low fat diet: Mean difference (95%CI): −0.56 (−1.01, −0.11) kg/m ↓ WC had no difference in MED vs. Low fat diet: Mean difference (95%CI): −0.89 (−1.96,0.18) cm	↓ SBP was greater in MED vs. Low fat diet: Mean difference (95%CI): −0.56 (−1.01, −0.11) mmHg ↓ DBP was greater in MED vs. Low fat diet: Mean difference (95%CI): −1.47 (−2.14, −0.81) mmHg ↓ TC was greater in MED vs. Low fat diet: Mean difference (95%CI): −7.35 (−10.32, −4.39) mg/dL ↓ LDL-C had no difference in MED vs. Low fat diet: Mean difference (95%CI): −3.34 (−7.27,0.58) mg/dL ↑ HDL-C had no difference in MED vs. Low fat diet: Mean difference (95%CI): 0.94 (−1.93,3.82) mg/dL ↓ hs-CRP was greater in MED vs. Low fat diet: Mean difference (95%CI): −0.97 (−1.49, −0.46) mg/dL ↓ FBG was greater in MED vs. Low fat diet: Mean difference (95%CI): −3.83 (−7.04, −0.62) mg/dL ↓ INS had no difference in MED vs. Low fat diet: Mean difference (95%CI): −1.06 (−2.94,0.81) µU/mL
Stendell-Hollis et al., 2013 USA	RCT 4 months follow-up 129 overweight women mean BMI 27.2 kg/m^2^ breastfeeding 73.6% mean postpartum: 17.5 weeks	MED (*n* = 53)	MyPyramid diet (*n*= 49)	Before and after changes of all variables were significant in all groups. ↓ Weight (MED: −2.31 vs. MyPyramid diet: −3.11 kg, no difference) ↓ BMI (MED: −0.85 vs. MyPyramid diet: −1.13 kg/m^2^, no difference) ↓ WC (MED: −3.47 vs. MyPyramid diet: −4.59 cm, no difference) ↓ hip (MED: −2.19 vs. MyPyramid diet: −2.90 cm, no difference) ↓ WHR (MED: −0.02 vs. MyPyramid diet: −0.02, no difference) ↓ WHR (MED: −1.19% vs. MyPyramid diet: −2.20%, no difference)	No significant before and after changes in two variables in two groups except TNF-α reduction in MyPyramid diet. ↓ IL6 (MED: −0.39 vs. MyPyramid diet: −0.03 pg/mL, no difference) ↓ TNF-α (MED: −0.89 vs. MyPyramid diet: −0.82 pg/mL, no difference
Rodriguez-Garcia et al., 2017 Spain	Open, single-blind study 3 months follow-up N = 115 women with MHO Age: 35–55 y BMI: 30–40 kg/m^2^	MED and physical exercise	Before intervention	↓ Significant weight reduction (Before vs. after: Mean 92.7 ± 13.8 vs. 86.5 ± 14.0 kg) ↓ Significant BMI reduction (Before vs. after: Mean 36.3 ± 4.7 vs. 33.8 ± 4.4 kg/m^2^) ↓ Significant WC reduction (Before vs. after: Mean 111.7 ± 11.1 vs. 106.0 ± 10.3 cm) ↓ Significant BMI reduction (Before vs. after: Mean 30.7 ± 6.0 vs. 28.7 ± 5.6 kg/m^2^)	↓ SBP (Before vs. after: Mean 114 ± 14 vs. 113 ± 13 mmHg, no difference) ↓ DBP (Before vs. after: Mean 76 ± 9 vs. 74 ± 10 mmHg, no difference) ↓ Significant TC reduction (Before vs. after: Mean 194.6 ± 28.2 vs. 181.1 ± 32.8 mg/dL) ↓ Significant LDL-C reduction (Before vs. after: Mean 114.5 ± 23.3 vs. 108.0 ± 25.6 mg/dL) ↓ Significant HDL-C increase (Before vs. after: Mean 56.5 ± 12.5 vs. 53.7 ± 12.5 mg/dL) VLDL (Before vs. after: Mean 10.7 ± 8.4 vs. 11.3 ± 10.0 mg/dL, no difference) ↓ Significant small dense LDL-C number reduction (Before vs. after: Mean 394.0 ± 84.2 vs. 378.9 ± 97.0)
Bajerska et al., 2018 Poland	Two-arm RCT 16 months follow-up Post-menopausal women	MED diet: 37% energy from total fat, 20% from MUFAs, 9% from PUFAs, 8% from SFAs, 18% from protein, and 45% energy from carbohydrates. Olive oil in every meal and 5–7 nuts/day	CED: Based on the recommendations of the NCEP and the AHA, (27% energy from total fat, 10% from MUFAs, 9% from PUFAs, 8% from SFAs, 18% from protein, and 55% energy from carbohydrate, dietary fiber from typical food of the central European region	Before and after changes of all variables were significant in all groups. ↓ Weight (MED: −7.7 vs. CED: −7.6 kg, no difference) ↓ WC (MED: −7.4 vs. CED: −7.4 cm, no difference) ↓ FM (MED: −6.7% vs. CED: −6.6%, no difference) ↓ FFM (MED: −1.1% vs. CED: −0.8%, no difference) ↓ FFM (MED: −0.25% vs. CED: −0.26%, no difference)	Before and after changes of all variables were significant in all groups. ↓ INS (MED: −3.5 vs. CED: −3.1 µU/mL, no difference) ↓ HOMA2-IR (MED: −0.46 vs. CED: −0.42, no difference) ↓ TC (MED: −15.5 vs. CED: −11.2 mg/dL, no difference) ↓ LDL-C (MED: −9.4 vs. CED: −4.9 mg/dL, no difference) ↓ HDL-C (MED: −0.1 vs. CED: −2.0 mg/dL, no difference) ↓ TG (MED: −33.9 vs. CED: −33.8 mg/dL, no difference) ↓ Hcy (MED: −0.7 vs. CED: −0.8 mg/dL, no difference) ↓ SBP (MED: −10.2 vs. CED: −10.4 mmHg, no difference) ↓ SBP (MED: −6.7 vs. CED: −8.1 mmHg, no difference)

M MDS: modified-Mediterranean Diet Score; BMI: Body mass index; WC: Waist circumference; RCT: Randomised clinical trial; MED: Mediterranean; MUFA: Monounsaturated fatty acid; PUFA: Saturated fatty acid; Saturated fatty acid; CED: Central European Diet; NCEP: National Cholesterol Education Program; AHA: American heart Association; FM: Fat mass; FFM: Fat free mass; INS: Insulin; TC: Total cholesterol; LDL-C: Low density lipoprotein cholesterol; HDL-C: High density lipoprotein cholesterol; TG: Triglycerides; Hcy: Hemosystein; SBP: Systolic blood pressure; DBP: Diastolic blood pressure; FBG: Fasting blood glucose; WHR: Waist to hip ratio; MHMD: Moderately hypo energetic Mediterranean diet; ** Denmark, France, Germany, Greece, Italy, The Netherlands, Norway, Spain, Sweden, and the United Kingdom.

**Table 3 nutrients-13-03504-t003:** Summary of studies with DASH diets for weight loss, obesity, cardiometabolic and cancer outcomes.

First Author, Year, Country	Design and Population	Intervention	Comparator/Control	Obesity Outcomes	Cardiometabolic and Cancer Outcomes
Asemi et al., 2014 Iran	RCT 8 weeks follow-up 48 overweight and with obesity with polycystic ovary syndrome 18–40 years BMI: ≥25 kg/m^2^	Calorie-restricted DASH diet (−350–700 kcal/day, according to BMI) *n* = 24	Calorie-restricted control diet (−350–700 kcal/day, according to BMI) *n* = 24	↓ Weight (−3.6 vs. −1.3 kg; *p* < 0.001); ↓ BMI (−1.3 vs. 0.4 kg/m^2^; *p* < 0.001); ↓ WC (−5.2 vs. −2.1 cm; *p* = 0.003); ↓ HC (−5.9 vs. −1 cm; *p* < 0.0001)	↓ serum insulin levels (−1.88 vs. 2.89 μIU/mL, *p* = 0.03); ↓ HOMA-IR score (−0.45 vs. 0.80; *p* = 0.01); ↓ serum hs-CRP levels (−763.29 vs. 665.95 ng/mL, *p* = 0.009) ↔ FPG, HOMA-B
Bertoia et al., 2014 United States	Cohort 3 years follow-up 93,122 postmenopausal women	Mediterranean diet score DASH diet score	Quintile cut-offs	↓ BMI (*p* < 0.01) ↓Quartile 3 and 4 vs. lowest quintile waist-to-hip ratio (*p* < 0.01)	↔ Sudden cardiac death
Soltani et al., 2016	Meta-analysis of RCT 8–52 weeks follow-up 13 articles 2292 overweight and adults with obesity BMI: ≥25 kg/m^2^	DASH diet	Usual/control diet/(2 articles with reduced-calorie diet and 1 counselling based on standard care)	↓ Weight (WMD = −1.45 kg; *p* = 0.082) ↓ BMI (WMD = −0.9 kg m^2^, 95%CI: −1.16, −0.64; *p* < 0.001)	
Foroozanfard et al., 2017 Iran	RCT 12-week follow-up 60 overweight or with obesity with polycystic ovary syndrome 18–40 years BMI: ≥25 kg/m^2^	Calorie-restricted DASH diet (−350–700 kcal/day, according to BMI) *n* = 30	Calorie-restricted control diet (−350–700 kcal/day, according to BMI) *n* = 30	↓ Weight (−4.3 kg; *p* = 0.01) ↓ BMI (−1.6 vs. 1.2 kg/m^2^, *p* = 0.02)	↓ AMH (−1.1 vs. 0.3 ng/mL, *p* = 0.01); ↓ insulin (−25.2 vs. −1.2 pmol/L, *p* = 0.02); ↓ HOMA-IR (−0.9 vs. −0.1; *p* = 0.02); ↓ HOMA-B (−16.4 vs. −1.0; *p* = 0.03); ↓ MDA levels (−0.5 vs. 0.2 µmol/L, *p* < 0.001); ↓ FAI (−0.03 vs. 0.06; *p* = 0.02); ↑ QUICKI (0.01 vs. −0.004; *p* = 0.02); ↑ SHBG (3.7 vs. −1.5 nmol; *p* = 0.01); ↑ NO levels (9.0 vs. 0.6 µmol/L, *p* < 0.001) ↔ Total testosterone, FSH, LH, 17-OH progesterone
Fulay et al., 2018 United States	Cohort Gestational period follow-up 1760 pregnant women	DASH diet DASH OMNI diet DASH + unsaturated fat intake supplemented	Quintile cut-offs	↔ GWG in normal weight women ↑ GWG among women with obesity before pregnancy (*p* ≤ 0.05)	↔ Hypertensive disorders, gestational diabetes

BMI: WC: Body Mass Index; waist circumference; GWG: gestational weight; FPG: fasting plasma glucose; HC: hip circumference; TGs: triglycerides; TAC: total antioxidant capacity; AMH: Anti-Müllerian hormone; SHBG: sex hormone-binding globulin; MDA: serum malondialdehyde level; WMD: weighted mean difference; * women’s results presented separately; ↔ no significant change, ↑ significant increase, ↓ significant reduction.

**Table 4 nutrients-13-03504-t004:** Summary of studies with intermittent fasting diet for weight loss, obesity and cardiometabolic outcomes.

First Author, Year, Country	Design and Population	Intervention	Comparator/Control	Obesity Outcomes	Cardiometabolic and Cancer Outcomes
Klempel et al., 2012 USA	RCT 10-week follow-up 46 women aged 35–65 y, BMI: 30–39.9 kg/m^2^	IFCR-L	IFCR-F	↓ Weight significantly greater for IFCR-L vs. IFCR-F (mean change in IFCR-L: 3.9 ± 1.4 kg (4.1 ± 1.5%) vs. IFCR-F: 2.5 ± 0.6 kg (2.6 ± 0.4%)) ↓ BMI significantly greater for IFCR-L vs. IFCR-F (mean change in IFCR-L: 1.3 ± 0.5 vs. IFCR-F: 0.8 ± 0.5 kg/m^2^) ↓ FM significantly greater for IFCR-L vs. IFCR-F (mean change in IFCR-L: 2.8 ± 1.2 vs. IFCR-F: 1.9 ± 0.7 kg/m^2^) ↓ Visceral fat significantly greater for IFCR-L vs. IFCR-F (mean change in IFCR-L: 0.7 ± 0.5 vs. IFCR-F: 0.3 ± 0.5 kg) FFM change had no difference in both groups	↓ TC significantly greater for IFCR-L vs. IFCR-F (mean change in IFCR-L: 19 ± 10% vs. IFCR-F: 8 ± 3%) ↓ LDL-C significantly greater for IFCR-L vs. IFCR-F (mean change in IFCR-L: 20 ± 9% vs. IFCR-F: 7 ± 4%) HDL-C had no difference in both groups ↓ Small dense LDL-C significantly greater for IFCR-L vs. IFCR-F (mean change in IFCR-L: 9 ± 4% vs. IFCR-F: 3 ± 1%) Heart rate had reduction in IFCR-L and increase in IFCR-F and difference was significant between two groups (−3 ± 4 vs. 3 ± 2) SBP, DBP, FBG, INS, CRP, Adeponectin and Leptin changes had no difference in both groups
Trepanowski et al., 2017 USA	RCT 6- and 12-month follow-up N = 100 adults with obesity, 84% women 18–64 y Mean BMI: 34 kg/m^2^	ADF: 25% of energy needs on fast days; 125% of energy needs on alternating “feast days”	DCR: 75% of energy needs every day Control: No-intervention	No significant difference between ADF and DCR ↓ Weight significantly greater for ADF vs. control at 6 and 12 months, respectively (mean difference: −6.8 (−9.1, −4.5) % and −6.0 (−8.5, −3.6)% ↓ FM significantly greater for ADF vs. control at 6 months, (mean difference: −4.2 (−6.6, −1.8) kg ↓ Visceral significantly greater for ADF vs. control at 6 and 12 months, respectively (mean difference: −0.4 (−0.7, −0.1) kg and −0.4 (−0.7, −0.1) kg	No significant difference between ADF and DCR ↓ HR significantly greater for ADF vs. control at 6 months, (mean difference: −5.8 (−11.3, −0.3) beats/min ↑ HDL-CR significantly greater for ADF vs. DCR at 6 months, (mean difference: 8.4 (1.9, 14.7) mg/dL ↓ TG significantly greater for ADF vs. control at 6 and 12 months, respectively (mean difference: −19.1 (−36.3, −1.8) and −24.4 (−43.5, −5.3) mg/dL ↓ INS significantly greater for ADF vs. control at 6 and 12 months, respectively (mean difference: −7.5 (−12.9, −2.0) and −5.9 (−11.7, −0.1) µIU/mL ↓ HOMA-IR significantly greater for ADF vs. control at 6 months, (mean difference: −2.49 (−4.22, −0.76) kg TC, LDL-C, FBG, SBP, DBP, hs-CRP, Hemocyctein had no significant changes at 6- and 12-months follow-up in 3 groups.
Trepanowski et al., 2018 USA	RCT 12- and 24-week follow-up N = 79, 83% women Overweight and adults with obesity aged 18–65 y BMI: 25–39.9	ADF: 25% of energy needs on fast days; 125% of energy needs on alternating “feast days”	DCR: 75% of energy needs every day Control: No-intervention	↓ Leptin: The ADF group and DCR group experienced greater reductions over time compared with the control group, but similar reductions compared to each other Adiponectin and resistin had no significant changes	↓ INS: The ADF group and DCR group experienced greater reductions over time compared to the control group, but similar reductions compared to each other ↓ HOMA-IR: The ADF group experienced greater reductions over time compared to the DCR and control groups
Beaulieu et al., 2019 USA	RCT 12-week follow-up N = 66 women Volunteer with obesity and overweight 18–55 y BMI: 25.0–34.9 kg/m^2^	IER diet (25% energy needs)	CER diet (75% energy needs)	↓ BMI, FM, FFM, fat percentage and WC significantly reduced in both groups, but no difference between groups	-
Panizza et al., 2019 USA	RCT 12 weeks follow up N = 60 volunteers aged 35–55 70% women BMI: 25–40 kg/m^2^, VAT ≥ 90 cm^2^ for men and ≥ 80 cm^2^ for women	IER + MED diet	DASH diet	↓ Weight significantly greater for IER + MED vs. DASH (mean change in IER + MED: 5.9 ± 0.7 vs. DASH: 3.3 ± 0.6 kg) ↓ BMI significantly greater for IER + MED vs. DASH (mean change in IER + MED: 2.2 ± 0.2 vs. DASH: 1.2 ± 0.2 kg/m^2^) ↓ WC significantly greater for IER + MED vs. DASH (mean change in IER + MED: 6.9 ± 0.8 vs. DASH: 4.5 ± 0.7 cm) ↓ Body fat significantly greater for IER + MED vs. DASH (mean change in IER + MED: 2.0 ± 0.4% vs. DASH: 0.8 ± 0.4%) ↓ FM significantly greater for IER + MED vs. DASH (mean change in IER + MED: 3.3 ± 0.4 vs. DASH: 1.6 ± 0.4 kg) ↓ VAT significantly greater for IER + MED vs. DASH (mean change in IER + MED: 22.6 ± 3.6 vs. DASH: 10.7 ± 3.5 cm^2^) ↓ SAT significantly greater for IER + MED vs. DASH (mean change in IER + MED: 48.2 ± 6.4 vs. DASH: 15.0 ± 6.1 cm^2^)	↓ TC significantly reduced only in IER + MED (mean change in IER + MED: 17.4 ± 6.4 and DASH: 9.1 ± 6.2 mg/dL, but no difference between two groups) ↓ LDL-C significantly reduced only in IER + MED (mean change in IER + MED: 14.0 ± 5.8 and DASH: 9.5 ± 5.8 mg/dL, but no difference between two groups) ↓ TG significantly greater for IER + MED vs. DASH (mean change in IER + MED: 24.8 ± 8.2 vs. DASH: 22.0 ± 7.9 mg/dL), but no difference between two groups ↓ SBP significantly greater for IER + MED vs. DASH (mean change in IER + MED: 9.0 ± 2.5 vs. DASH: 5.7 ± 2.4 mmHg), but no difference between two groups ↓ DBP significantly greater for IER + MED vs. DASH (mean change in IER + MED: 6.7 ± 1.5 vs. DASH: 3.4 ± 1.4 mmHg), but no difference between two groups ↓ INS significantly greater for IER + MED vs. DASH (mean change in IER + MED: 5.1 ± 1.2 vs. DASH: 2.5 ± 1.7 mU/L), but no difference between two groups ↓ AST significantly reduced only in IER + MED (mean change in IER + MED: 5.7 ± 2.2 and DASH: 1.6 ± 2.1 mg/dL, but no difference between two groups) ↓ FBG reduced non significantly in IER + MED and DASH (mean change in IER + MED: 2.1 ± 2.4 and DASH: 2.4 ± 2.3 mg/dL, but no difference between two groups)

RCT: Randomized clinical trial; BMI: Body mass index; IFCR-L: Intermittent fasting calorie restriction-liquid diet; IFCR-F: Intermittent fasting calorie restriction-food diet; FM: fat mass; FFM: Fat free mass; WC: Waist circumference; TC: Total cholesterol; LDL-C: Low density lipoprotein cholesterol; HDL-C: High density lipoprotein cholesterol; SBP: Systolic blood pressure; DBP: Diastolic blood pressure; FBG: Fasting blood glucose; INS: Insulin; CRP: C-reactive protein; CER: Continuous energy restriction; IER: Intermittent energy restriction; VAT: Visceral adipose tissue; MED: Mediterranean; DASH: Dietary Approach to Stop Hypertension; SAT: Subcutaneous adipose tissue; TG: triglyceride; AST: Aspartate trasaminase; ALT: Alanine trasaminase; ADF: Alternate day fasting; DCR: Daily calorie restriction; HR: heart rate.

**Table 5 nutrients-13-03504-t005:** Summary of studies with general healthy diet and/or food reduction healthy diets for weight loss, obesity, cardiometabolic and cancer outcomes.

First Author, Year, Country	Design and Population	Intervention	Comparator/Control	Obesity Outcomes	Cardiometabolic and Cancer Outcomes
Maslova et al., 2015 Denmark	Cohort Danish National Birth Cohort 20–24 weeks follow-up 46.262 pregnant women	Protein: carbohydrate ratio and added sugar	Quintile cut-offs	Protein: carbohydrate ratio: ↓ GWG (−16 g/week; <0.0001) Added sugar: ↑ GWG (34 g; *p* < 0.0001)	-
Renault et al., 2015 Denmark	RCT 342 pregnant women with obesity	Baseline highest quartile of added sugars foods Mediterranean-style hypocaloric diet (5000–7000 kJ) *n* = 114 (physical activity + dietary) *n* = 110 (physical activity)	Quartile cut-offs (baseline data) *n* = 118	Baseline added sugar ≥2/day: ↑ GWG (5.4 kg greater than < 1 week intake; *p* = 0.02)	-
Flynn et al., 2016 United Kingdom	RCT 28 weeks follow-up 1023 pregnant women with obesity	Behavioral intervention of diet (healthier pattern of eating) and physical activity advice Behavioral intervention: restricting the consumption of sugar-sweetened beverages, including fruit juice, and use low fat dairy products and replace fatty meats and meat products with lean meat or fish. *n* = 519	Quartile cut-offs (baseline data) *n* = 504	-	Baseline African/Caribbean-↑ Gestational diabetes (OR = 2.46) and Baseline Processed-↑ Gestational diabetes (OR = 2.05)
Stang et al., 2016	Position of the Academy of Nutrition and Dietetics Women of reproductive age with obesity (15–49 years)	Nutrition education and nutritional health care (lifestyle counselling and balanced diet calculated by nutritionist)	-	↓ GWG ↓ BMI ↑ postpartum weight loss ↓ postpartum weight retention	↓ gestational hypertension ↓ gestational diabetes ↓ pre-eclampsia
Casas et al., 2020	Systematic review 39 studies 681,383	Sugary food consumption and processed foods	-	Simple sugars and processed foods:-↑ GWG	Simple sugars and processed foods: ↑ Gestational diabetes, and ↑ Gestational hypertension
Garmendia et al., 2020 Chile	RCT 4631 pregnant women	Nutritional health care standards and practices at the primary health care 4 phases: 1) training of professionals on nutritional recommendations. 2) counselling of pregnant women on diet and physical activity;3) offer of a PA program implemented; and 4) adequate referral to primary health care centres dietitians *n*= 2565	Routine care *n* = 2066	↓ GWG general (11.3 vs. 11.9 kg; *p* = 0.003) ↓ GWG in pregestational women with obesity (8.6 vs.9.7 kg; *p* = 0.014)	↔ glucose concentration and Gestational diabetes
Hutchesson et al., 2020 (1998–2018)	Systematic review of RCT and systematic reviews 90 studies 26,750 women of reproductive age 15–44 years	Behavioral interventions (physical activity and sedentary and/or dietary behaviors	-	↑ weight loss ↓ excessive GWG ↓ postpartum weight retention	-

HEI: Healthy Eating Index; AHEI: Alternative Healthy Eating Index; BMI: body mass index; WC: waist circumference; * women’s results presented separately; ↔ no significant change, ↑ significant increase, ↓ significant reduction.

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
