# Peer review of "Which Diets Are Effective in Reducing Cardiovascular and Cancer Risk in Women with Obesity? An Integrative Review"

_nutrients, 2021, doi:10.3390/nu13103504_

Round 1

Reviewer 1 Report

However, there are some controversies on the effect of MED on 240 the secondary prevention of CVD 70

Please correct the typo

Overall, women have a higher prevalence of anxiety and depression

Please add a reference.

How is Brazillian diet helpful for men and obese children. Please write some lines on this .

In a meta-analysis of 13 randomized controlled clinical trials which included 2,292 overweight and adults with obesity, only  two studies presented women's data separately (n = 159).

Please mention two  studies.

Some studies have been conducted only with women.

Mention the studies.

 Four randomized clinical  trials have assessed the consumption of calorie-restricted DASH diet compared to a calorie-restricted control diet in overweight and women with obesity with polycystic ovary syndrome for 8-12 weeks.

Mention 4 studies.

The authors have not mentioned the consumption of linoleic acid in light of weight gain. Please cite the following papers in this context.

Dietary Linoleic Acid Elevates Endogenous 2-AG and Anandamide and Induces Obesity, Alvheim et.al., 2012, Obesity.

Linoleic Acid in Diets of Mice Increases Total Endocannabinoid Levels in Bowel and Liver: Modification by Dietary Glucose, Ghosh et.al., 2019,Obesity Science and Practice.

Author Response

Reviewer #1:

Comments and Suggestions for Authors

However, there are some controversies on the effect of MED on the secondary prevention of CVD (70).

Please correct the typo

Response: Thank you. We have corrected the typo.

Overall, women have a higher prevalence of anxiety and depression

Please add a reference.

Response: Thank you. We have included a reference to support this statement.

How is Brazilian diet helpful for men and obese children. Please write some lines on this.

Response: Thank you for your comment. However, there is no evidence to date on men or obese children and the Brazilian diet.

In a meta-analysis of 13 randomized controlled clinical trials which included 2,292 overweight and adults with obesity, only two studies presented women's data separately (n = 159).

Please mention two studies.

Response: Thank you. We have included the reference for this meta-analysis and for studies that presented women’s data separately. Furthermore, we have corrected the number of articles that have included only women or that presented data for women separately i.e. three instead of two.

Some studies have been conducted only with women.

Mention the studies.

Response: We have added these references.

Four randomized clinical trials have assessed the consumption of calorie-restricted DASH diet compared to a calorie-restricted control diet in overweight and women with obesity with polycystic ovary syndrome for 8-12 weeks.

Mention 4 studies.

Response: Thank you. We have included these references.

The authors have not mentioned the consumption of linoleic acid in light of weight gain. Please cite the following papers in this context.

Dietary Linoleic Acid Elevates Endogenous 2-AG and Anandamide and Induces Obesity, Alvheim et.al., 2012, Obesity.

Linoleic Acid in Diets of Mice Increases Total Endocannabinoid Levels in Bowel and Liver: Modification by Dietary Glucose, Ghosh et.al., 2019, Obesity Science and Practice.

Response: Thank you for your suggestion. However, our main objective was to focus on diets. Therefore, have decided not to include specific foods or functional foods in this manuscript since it would require an additional in-depth review of the scientific literature on this topic. In addition, we have decided not to include evidence from animal models.

Reviewer 2 Report

Article of great importance due to the importance of diet in the prevention of diseases in general and more specifically of cardiovascular diseases and cancer. It is a review study with which biases can be neutralized and we have a more global vision of the subject.

It is a study with an adequate introduction, material and correctly planned method, highlighting the great variety of diets analyzed. The results are clearly expressed despite being very numerous, but with the help of the tables that summarize the analyzed articles very well, everything is clear and synthetic. The discussion is adequate.

In summary, I find it a very interesting article that highlights the great importance of having a proper diet in the prevention of cancer and cardiovascular diseases.

Author Response

Reviewer #2:

Comments and Suggestions for Authors:

Article of great importance due to the importance of diet in the prevention of diseases in general and more specifically of cardiovascular diseases and cancer. It is a review study with which biases can be neutralized and we have a more global vision of the subject.

It is a study with an adequate introduction, material and correctly planned method, highlighting the great variety of diets analyzed. The results are clearly expressed despite being very numerous, but with the help of the tables that summarize the analyzed articles very well, everything is clear and synthetic. The discussion is adequate.

In summary, I find it a very interesting article that highlights the great importance of having a proper diet in the prevention of cancer and cardiovascular diseases.

Response: Thank you for your in-depth review and your feedback. We also appreciate your kind words about the importance of our study.
